**Data Availability Statement:** The paper and its Supporting Information files contain all necessary data. Our data was uploaded to a public repository

# Persistent expression of activation markers on *Mycobacterium tuberculosis*-specific CD4 T cells in smear negative TB patients

Ahmed Esmael[1,2,3]*, Adane Mihret[2,3], Tamrat Abebe[2], Daniel Mussa[3], Sebsibe Neway[3], Joel Ernst[4], Jyothi Rengarajan[5], Liya Wassie[3]☯, Rawleigh Howe[3]☯

**1** Department of Medical Laboratory Science, College of Health Sciences, Debre Markos University, Addis Ababa, Ethiopia, **2** Department of Microbiology, Immunology and Parasitology, College of Health Sciences, Addis Ababa University, Addis Ababa, Ethiopia, **3** Armauer Hansen Research Institute, Addis Ababa, Ethiopia, **4** Division of Experimental Medicine, University of California San Francisco, San Francisco, California, United States of America, **5** Division of Infectious Diseases and Emory Vaccine Center, Department of Medicine, Emory University School of Medicine, Emory University, Atlanta, Georgia, United States of America

☯ These authors contributed equally to this work.
* esmaelahmed8@gmail.com

## Abstract

### Background

T cell activation (HLA-DR, CD-38), proliferation (KI-67), and functional (IFN-γ, TNF-α) markers have recently been shown to be useful in predicting and monitoring anti-TB responses in smear positive TB, but previous research did not characterize the activation and proliferation profiles after therapy of smear negative TB.

### Methodology

In this study, we used polychromatic flow cytometry to assess selected PPD-specific T cell markers using fresh PBMC of smear negative and positive pulmonary tuberculosis (PTB) patients, recruited from health facilities in Addis Ababa.

### Result

Levels of activation (HLA-DR, CD38) and proliferation (Ki-67) among total unstimulated CD4 T cells decreased significantly after therapy, particularly at month 6. Similarly, levels of PPD-specific T cell activation markers (HLA-DR, CD-38) were significantly lower in smear positive PTB patients following treatment, whereas a consistent decline in these markers was less apparent among smear negative PTB patients at the sixth month.

### Conclusion

After six months of standard anti-TB therapy, persistent levels of activation of HLA-DR and CD-38 from PPD specific CD4+T cells in this study could indicate that those markers have little value in monitoring and predicting anti-TB treatment response in smear negative pulmonary TB patients in Ethiopian context.

and may be found at https://figshare.com/account/home, doi 10.6084/m9.figshare.19753798.

**Funding:** This work was supported in part by the NIH/Fogarty International Center Global Infectious Diseases grant D43TW009127 and core fund from Armauer Hansen Research Institute (AHRI). The funders had no role in study design, data collection and analysis, decision to publish, or preparation of the manuscript.

**Competing interests:** The authors have declared that no competing interests exist.

## Introduction

Tuberculosis is a substantial public health threat, which has also worsened as a result of the public health challenges associated with COVID-19 [1]. To address this issue, the World Health Organization (WHO) has adopted Directly Observed Treatment Short-course (DOTS) as a cost-effective TB control strategy, which consists of a combination of antibiotics (i.e. isoniazid, rifampicin, pyrazinamide, and ethambutol) used to treat drug-susceptible *M.tuberculosis* isolates [1].

Tools to monitor and predict anti-TB treatment outcomes in patients have shown to have poor sensitivity and specificity. Standard Ziehl-Neelsen smear microscopy, for example, has reduced sensitivity after two months of anti-TB drug treatment due to poor sputum quality [2–6]. Another concern is that smear microscopy and recently adopted molecular diagnostic methods indicated non-viable bacilli during anti-TB drug therapy monitoring [7–11]. Furthermore, the high infrastructure costs and long turnaround time of sputum culture limit its use as a reliable monitoring and therapy prediction tool for anti-TB drugs [5].

During *M. tuberculosis* infection, the cell mediated immune response, which is mediated by the T cell, is a crucial component in determining whether the infection is latent or progresses to active illness [12]. It is well known that CD4+ T cells play the most important role in *M. tuberculosis* infection; however, the role of CD8+ T cells during this bacterial infection has recently become clearer than in the past, beginning with peptide recognition that binds to major histocompatibility complex I and can secrete IL-12, IL-17, IL-2, IL-10, IFN-γ, TNF- α and TGF-β -, just like CD4+T cells. Furthermore, CD8+T cells use granulysin, perforin, and granzymes to cytolytically destroy infected cells, or they use Fas-Fas ligand interaction to cause apoptosis [12–14]. Recently, researchers looked at the frequency, phenotypic, and functional properties of CD4+T cells and CD8+ T cells in TB patients during anti-TB treatment and found inconsistent findings, with some reporting increases, decreases, and no change when compared to active TB patients before starting anti-TB drugs [15, 16].

Peptides of varying sensitivity and specificity (e.g., PPD, ESAT-6, CFP-10, M.tb lysate, Rv2628, Rv1733, Rv2031, Rv340, and Ag85) could be used during the T cell stimulation experiment. For instance, studies showed T cell response to ESAT-6-CFP-10 pool had a sensitivity and specificity of 67 and 100 percent, respectively, but T cell response to PPD had a sensitivity and specificity of 100 and 72 percent, respectively [17, 18].

Currently blood-based studies have demonstrated the utility of T cell activation (HLA-DR, CD-38), proliferation (KI-67), and functional (IFN-γ, TNF- α) markers in predicting and monitoring anti-TB drug responses. However, levels of activation, proliferation, and functional biomarkers varied across studies, with some reporting a decrease [19–23], others an increase [24–26], or no change after starting standard anti-TB drugs treatment [26]. For instance, Feruglio *et al* tried to investigate biomarkers to monitor effective anti-TB treatment and found the level of expression of HLA-DR/CD38 and PD-1/CD38 on both CD4(+) and CD8(+) T cells were significantly reduced after anti-TB treatment whereas the level of proliferation and cytokine production did not differ after anti-TB treatment [26]. Moreover, Ahmed *et al*. showed that At nine weeks of anti- TB treatment, the frequency of T cell activation and proliferation markers (CD38, HLA-DR, Ki67) were significantly reduced (p< 0.0001) whereas the frequency of CD4+ T was not changed. This study aslo found the reduction of activation markers expression related with culture conversion which indicate bacterial load reduction [20].

Another study by Vickers *et al* from whole blood stimulated with ESAT-6/CFP-10 and PPD found following ESAT-6/CFP-10 stimulation a significantly higher frequency of CD38 and HAL-DR on CD4+CD27+ cells at 2 months of anti-TB treatment were observed when

compared to baseline [24]. Also, study by Priyanto *et al* from whole blood stimulated with PPD found following PPD stimulation total PPD specific CD4+T cells did not show any statistical difference before anti-TB treatment and at 8 weeks of anti-TB treatment whereas the expression of CD38+ HLA DR+ on these cells were significantly lower at 8 weeks of anti-TB treatment initiation. However, the decrease was not specific for Interferon-gamma (IFNg) and IL-2 and conclude measurement of CD38+ HLA DR+ could be potential biomarkers to monitor anti-TB treatment success [21].

Furthermore, there have been few reports of useful markers in the monitoring of smear negative TB. In the present study, we used polychromatic flow cytometry to evaluate the monitoring and prediction potential of antigen-specific T cell activation, proliferation, and functional blood-based biomarkers for smear negative anti-TB drug treatment response in Ethiopia, a resource-constrained high TB endemic setting.

## Methodology

### Study setting

From August 2020 to July 2021, an institutional-based longitudinal cohort was conducted in selected health centers across Addis Ababa. Smear negative pulmonary TB patients were the main study subjects, whereas smear positive pulmonary TB patients and apparently healthy study participants served as comparison groups. All study participants were adults who were Human Immunodeficiency virus (HIV)-negative and had never been treated with anti-TB drugs. This study is a follow-up to a prior TB diagnostic biomarker research, which focused on smear negative PTB. Study participants were evaluated at baseline in the prior study, and the current study focused on two- and six-month follow-ups after starting medication. 115 study participants (smear negative TB-29, latent TB-30, apparently healthy-22, and smear positive TB-34) at baseline before treatment initiation, 56 participants (smear negative-25, smear positive-31) at second months, and 50 participants (smear negative-25, smear positive-25) at six months cohort from seven selected governmental health facilities in Addis Ababa, Ethiopia were included in this follow-up cohort [27].

We also operationally defined pulmonary TB patients as smear positive (Acid Fast Bacilli smear microscopy positive, and confirmed with Löwenstein–Jensen (LJ)/Mycobacterium growth indicator tube (MGIT) culture or RD9 polymerase chain reaction (PCR) positive) and smear negative (Acid Fast Bacilli smear microscopy negative, gene expert negative, and LJ/MGIT or RD9 PCR positive [27]. Furthermore, apparently healthy study participants were defined as those who did not have any signs or symptoms of TB and were Quantiferon TB Gold Plus (QFT) positive or negative.

### Cell isolation and flow cytometry

For flow cytometry analysis, 20 ml of heparinized whole blood was collected from the selected health centers and transported on ambient temperature to the AHRI laboratory for further laboratory analysis. In this study standardized protocol was adopted to separate mononuclear cells from heparinized whole blood using ficoll-hypaque density centrifugation technique [28]. Fresh mononuclear cells (1–2 million per well) were stimulated for 18 hours with purified protein derivatives (PPD, 10ug/ml) or phytohemagglutinin (PHA,5ug/ml). Brefeldin A (BD, USA) was added at 30 min for PHA and 2 hr for PPD during the 18 hours of stimulation period. Cells were harvested from replicate culture plates and stained with cocktail of surface monoclonal antibodies containing: 2.5ul CD8-APC-Cy7 (BD, USA), 2.5 ul HLA-DR-PE-Cy7 (BD, USA), 2.5ul CD38-BV421 (BD, USA), and 2.5 ul CD4-BV510 (BD, USA), followed with 30 minute incubation, washing with FACS buffer and cells permeabilized with Cytofix/perm

(BD, USA). Then, cells stained with a cocktail of intracellular monoclonal antibodies: 5ul TNF-α-APC (BD, USA), 5ul IFN-γ-FITC (BD, USA) and 5ul Ki-67-PerCP-C$^y$5.5 (BD, USA), followed with 30 minutes incubation, washed with Perm Wash (BD, USA) and fixed with 2% paraformaldehyde (PFA).After 30 minutes fixation with 2% PFA, cells were washed and resuspended with 500ul of FACS buffer until acquisition. Finally flow cytometry data acquisition were done with BD FACSCanto II using FACSDIVA software.

In addition, the Quantiferon TB Gold Plus technique (QIAGEN, Germany, optical density 450nm and 620nm, Analysis Software softmax$^R$pro7.013) was used to further classify apparently healthy study participants as QFT positive or QFT negative.

Further more productive sputum (5-10ml) was collected, digested and processed with N-acetyl L-cysteine- sodium hydro oxide method and inoculated on LJ/ MGIT $^{TM}$ 960 media in a SAFE FAST classic level 3 biosafety cabinet. Moreover, DNA extraction was done using DNA extraction kit (69504 and 69506) and final *M.tuberculosis* confirmation was done using RD9 PCR (RD-9 REV-RTPCR– 5`−CACTGCGGTCGCCATTG−3,TM-57-60OC, GC: 64.7%, 17 mer, RD9- FW-RTPCR- 5−TGCGGGCGGACAACTC−3,TM- 56 - 86OC,GC = 68.75%, 16mer, Eurofins genomics). RH37V and RNase free water were positive and negative controls, respectively.

## Data analysis

All sociodemographic and clinical data were cleaned and double-entered in an AHRI database management unit database. Flow cytometry data were analyzed with FlowJo analytical software (version 9.9.6). GraphPad Prism (Version 6.0 Software, California, USA) was used to analyze our data and to display in graphs. Stimulation indices was determined by dividing the frequency of cytokine producing cells in the presence of PPD stimulation with that in the absence of PPD stimulation. The proportion of IFN-γ+ and TNF-α+ producing CD4+ T cells was always at least two-fold higher in the PPD stimulated samples than the unstimulated (negative control) sample response. Data was analyzed using non-parametric Wilcoxon matched-paired rank test. The PHA was included as positive control. A p value less than 0.05 was considered statistically significant.

## Ethical considerations

The study was conducted according to the Declaration of Helsinki [29] and the protocol was approved by the Addis Ababa University-College of Health Sciences institutional review board and the AHRI/ALERT Ethics Review Committee. All participants gave written informed consent before enrolled into the study.

## Result

### Flow cytometric gating strategy

Lymphocyte populations were selected by forward and side scatter, doublets excluded and CD4+ cells gated (Fig 1). Finally, the expression of activation (HLA-DR, CD-38), proliferation (Ki-67), and cytokine (TNF-α, IFN-γ) markers on CD4$^+$ were defined as depicted.

### Changes in the cytokine markers profile from PPD specific CD4 T cells among smear negative and smear positive PTB patients

PBMC were stimulated with or without PPD antigens and assessed for intracytoplasmic cytokines. Stimulation indices (SI) represent the ratio of frequencies with PPD to that without PPD. Among smear negative PTB patients, the overall magnitude of PPD specific IFN-γ$^+$CD4$^+$ T cells and TNF-α$^+$CD4$^+$T cells fluctuated between baseline and the 6$^{th}$ month of

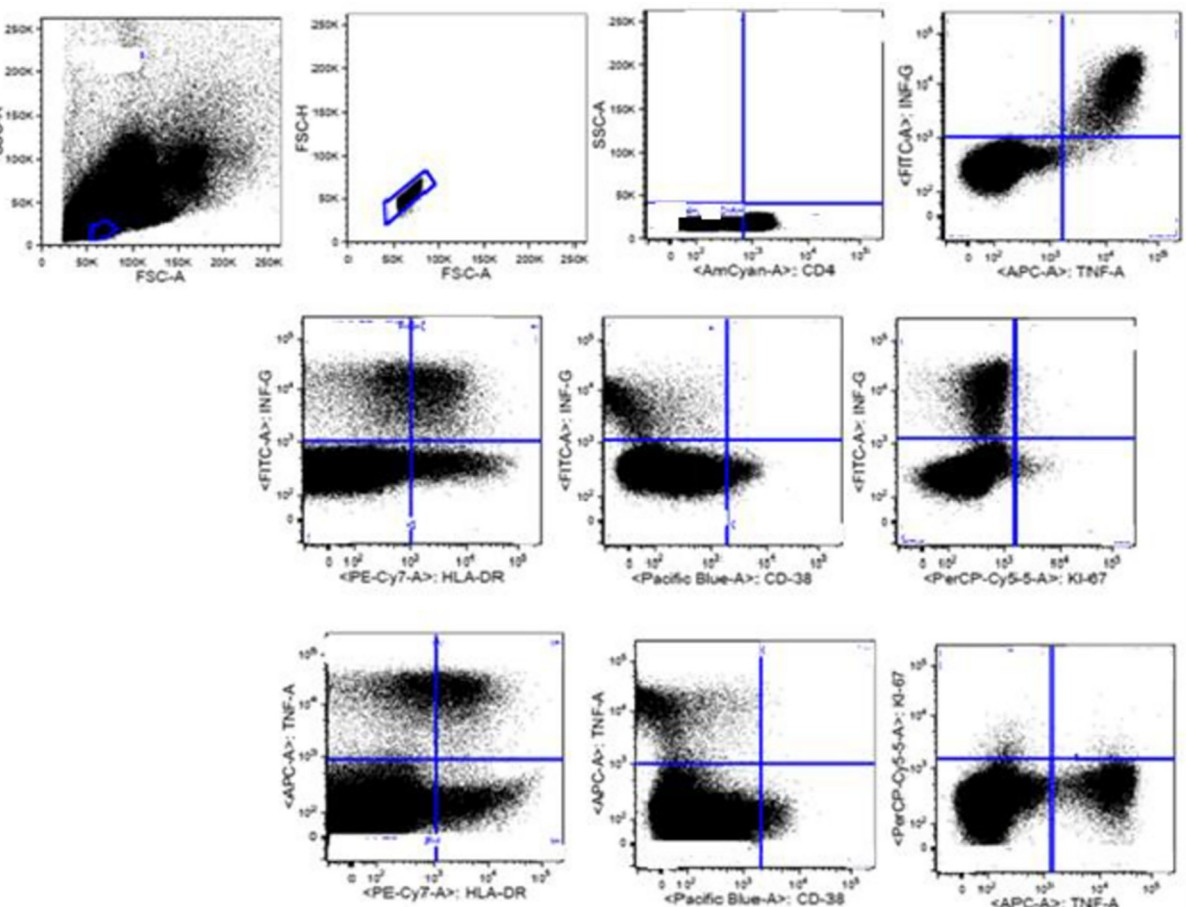

**Fig 1. Flow cytometry gating strategy for the TB biomarkers study in selected health centers, Addis Ababa, Ethiopia from August 2020-July 2021.**

therapy with a notable drop in frequency among IFN-γ⁺ TNF-α⁺ co-producing cells with therapy (Fig 2A). Among smear positive patients, a drop in frequency with therapy of IFN-γ⁺ and/or TNF-α⁺ PPD-specific CD4 T cells was more consistently observed after therapy (Fig 2B).

### Changes in the activation and proliferation markers profile from PPD specific CD4 T cells among smear negative and smear positive PTB patients

To determine the activation profile of PPD-specific cytokine producing cells, we assessed co-expression of HLA-DR, CD38, or Ki-67 among IFN-γ⁺ or TNF-α⁺ CD4+ T cells. As depicted in Fig 3, among smear negative patients, PPD specific activation or proliferation and cytokine co-expressing cells show fluctuations between different time points, in some cases reaching statistical significance. For example, HLA-DR⁺IFN-γ⁺, and CD-38⁺TNF-α⁺ levels were significantly reduced from baseline to month 2. However, the latter subset remained low at month 6, the former subset increased from month 2 to 6. Moreover, Ki-67+ TNF α or IFN- γ coproducing cells actually increased slightly from month 2 to month 6. Collectively, these results suggest that there may have been a modest decrease in activation or proliferation markers among PPD specific cytokine producing CD4 T cells, but that patterns were not consistently observed over the indicted follow up period (Fig 3A and 3B).

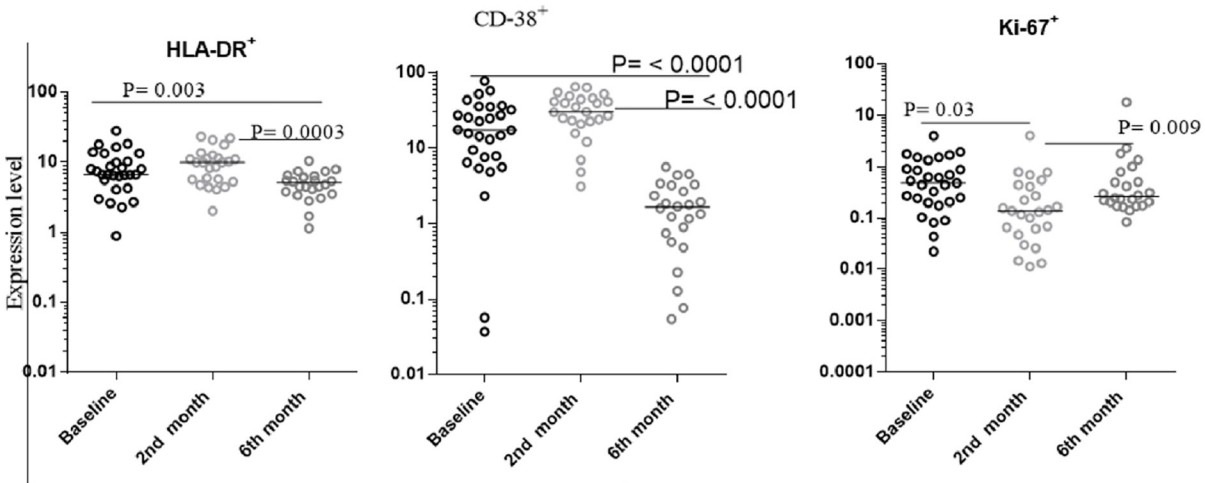

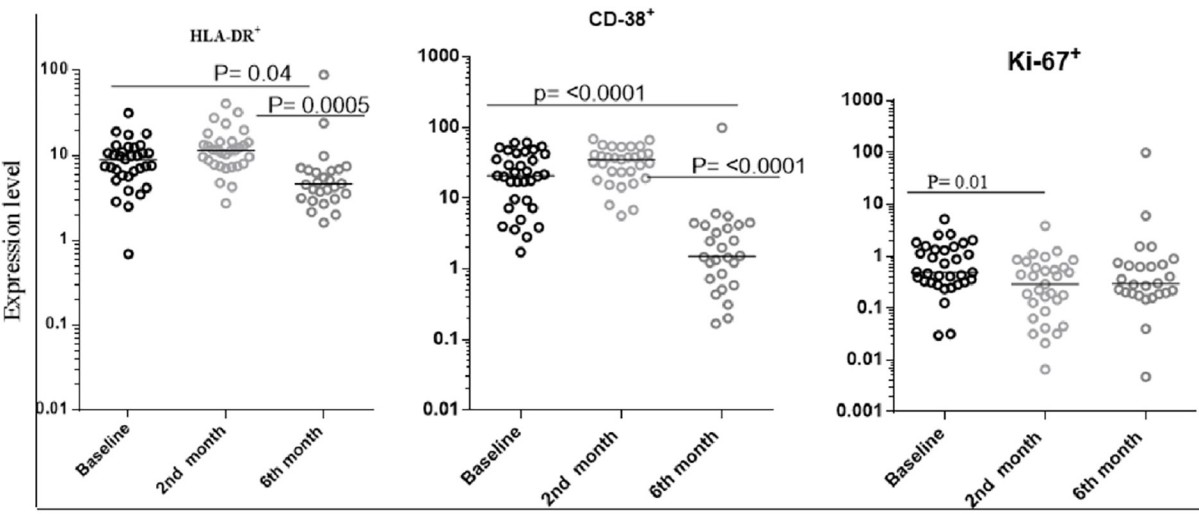

**Fig 2. The level of IFN-γ⁺ and TNF-α⁺ responses from PPD specific CD4+ T cells during the standard anti-TB treatment follow up time period.** A:—The level of IFN-γ⁺ and TNF-α⁺ responses among smear negative PTB patients. B:—The level of IFN-γ⁺ and TNF-α⁺ among smear positive PTB patients. Cytokine evaluation was performed after overnight stimulation with PPD or control cultures. The bars within each plot indicate the median value for each marker and time point. Data analysis was done using non-parametric Wilcoxon matched-paired rank test with significant p values indicated. Stimulation indices (SI) represent the ratio of frequencies in the presence of PPD to that of the absence of PPD. Baseline represents values prior to initiation of standard anti-TB drugs. 2nd and 6th month refers to time after therapy initiation.

Among smear positive patients, a more consistent decrease in HLA-DR or CD38 activation marker expression on IFN-γ or TNF-α co-producing PPD specific cells was observed. This was particularly apparent at the 6-month time point though also seen at the 2-month time

A

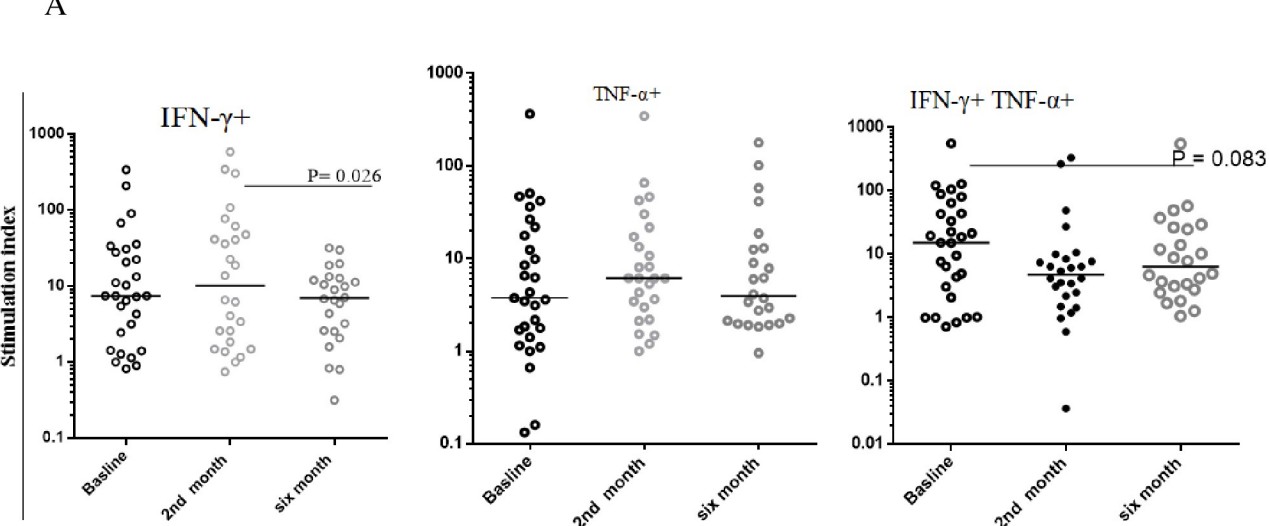

B

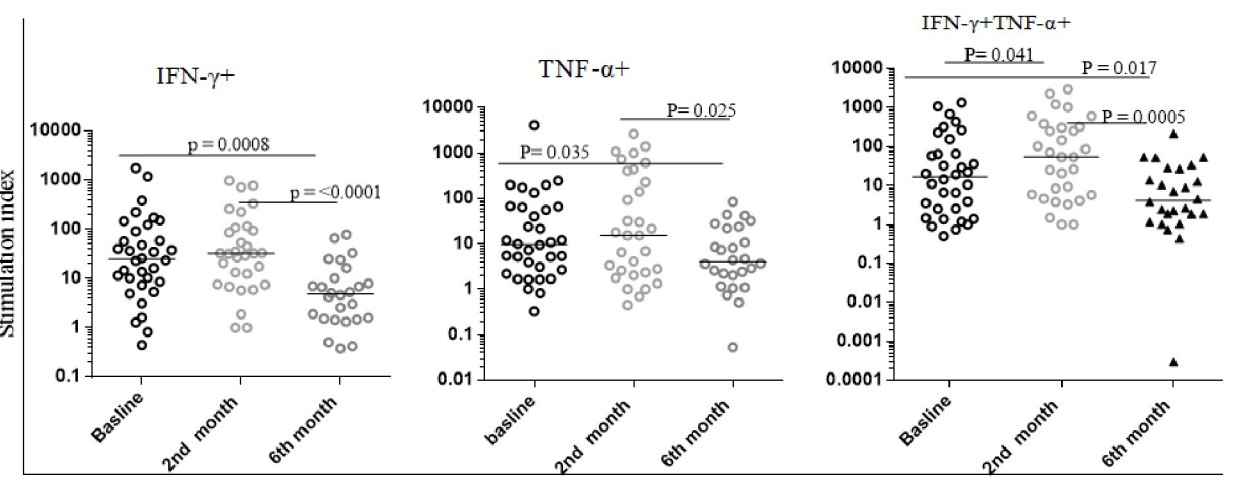

**Fig 3. Level of expression of activation and proliferation markers from PPD specific IFN-γ⁺CD4⁺T cells and TNF-α⁺ CD4⁺ T cells during the standard anti-TB treatment follow up using fresh PBMC samples.** A:—the level of activation and proliferation markers expression from IFN-γ⁺CD4⁺T cells among smear negative PTB patients. B:—the level of activation and proliferation markers expression from TNF-α⁺ CD4⁺ T cells among smear negative PTB patients. C:—the level of activation and proliferation markers expression from IFN-γ⁺CD4⁺T cells among smear positive PTB patients. D:—the level of activation and proliferation markers expression from TNF-α⁺ CD4⁺ T cells among smear positive PTB patients. Stimulated indices were calculated as the frequency of marker positive/cytokine positive cells cultured in the presence of PPD divided by that in the absence of PPD. The bars within each plot indicate the median value for each marker and time point. Data analysis was done using non-parametric Wilcoxon matched-paired rank test with significant p values indicated. Baseline represents values prior to initiation of standard anti-TB drugs. 2nd and 6th month refers to time after therapy initiation.

point among CD38+ TNF-α⁺ cells. There was also a modest drop in Ki-67+ IFN-γ cell at the 6-month time point. Collectively, these data indicate that there was a significant decrease in activation or proliferation markers among PPD-specific cytokine producing cells among

smear positive TB patients, and this decrease was more apparent than in smear negative TB cases (Fig 3C and 3D).

## Changes in the activation and proliferation markers profile from unstimulated CD4 T cells among smear negative and smear positive PTB patients

Frequencies of activation (HLA-DR and CD38) and proliferation (Ki-67) markers were assessed on CD4 T cells at baseline, 2 and 6 months after therapy commencement among smear negative and smear positive patients. HLA-DR and CD-38 expression on CD4$^+$ T cells among smear negative and positive PTB patients was considerably lower than the baseline and second month of therapy. However, a uniform drop was not seen for the Ki67 proliferation biomarker for both smear negative and positive PTB patients (Fig 4A and 4B).

A:—The level of activation and proliferation markers expression on CD4+T cells among smear negative PTB patients. B:- The level of activation and proliferation markers expression on CD4+T cells among smear positive PTB patients. The bars within each plot indicate the median value for each marker and time point. Data analysis was done using non-parametric Wilcoxon matched-paired rank test with significant p values indicated. Baseline represents values prior to initiation of standard anti-TB drugs. 2$^{nd}$ and 6$^{th}$ month refers to time after therapy initiation.

## Comparison of clinically cured smear positive and negative PTB patients with the apparently healthy comparators

Activation biomarkers were found to be considerably higher in smear positive and negative PTB patients at baseline when compared to QFT positive apparently healthy study participants. However, the level of activation markers on cytokine expressing smear positive and

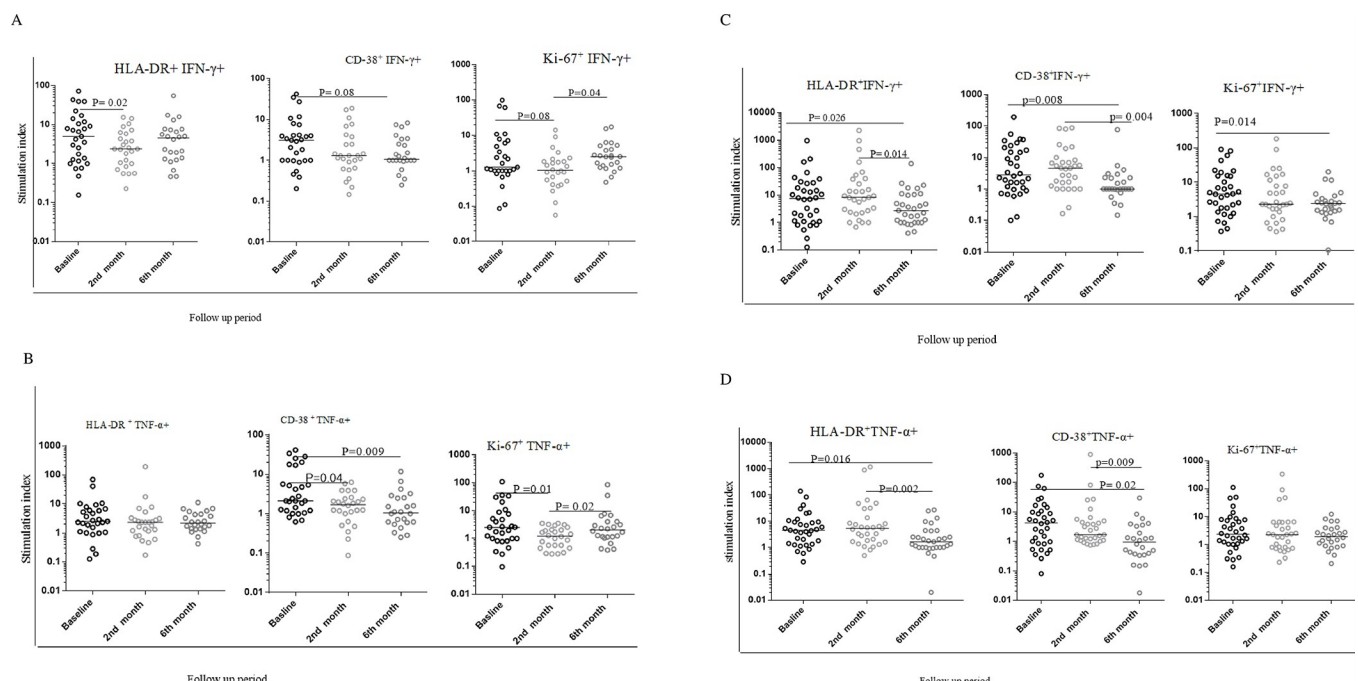

**Fig 4. The level of activation and proliferation marker expression on CD4+ T cells from unstimulated PBMC samples in TB biomarkers study cohort.**

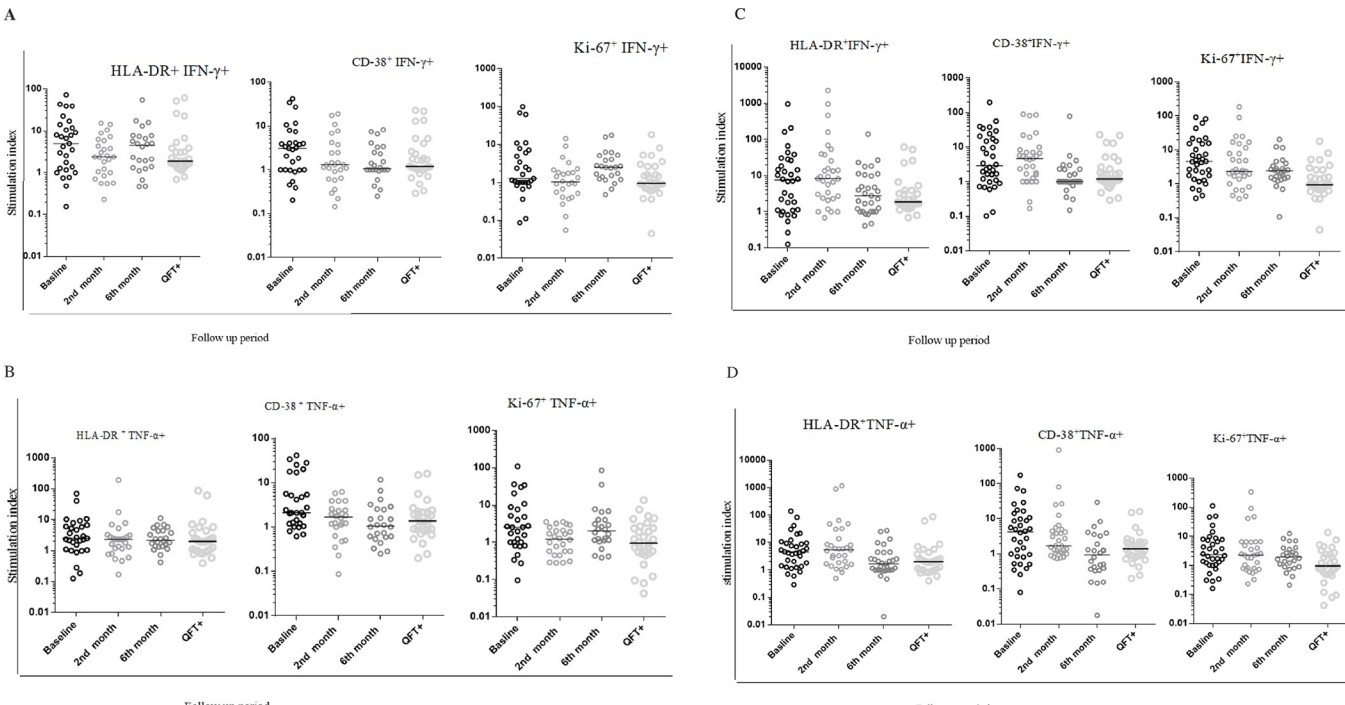

**Fig 5. Level of expression of activation and proliferation markers from PPD specific IFN-γ⁺CD4⁺T cells and TNF-α⁺ CD4⁺ T cells in clinically resolved (6 month's time point after therapy initiation) PTB patients and QFT positive healthy controls.** Refer to Fig 4 for further details. A:—the level of activation and proliferation markers expression from IFN-γ⁺CD4⁺T cells between smear negative PTB patients and QFT positive apparently healthy study participants. B:—the level of activation and proliferation markers expression from TNF-α⁺ CD4⁺ T cells between smear negative PTB patients and QFT positive apparently healthy study participants. C:—the level of activation and proliferation markers expression from IFN-γ⁺CD4⁺T cells between smear positive PTB patients and QFT positive apparently healthy study participants. D:—the level of activation and proliferation markers expression from TNF-α⁺ CD4⁺ T cells between smear positive PTB patients and QFT positive apparently healthy study participants. Stimulated indices were calculated as the frequency of marker positive/cytokine positive cells cultured in the presence of PPD divided by that in the absence of PPD. The bars within each plot indicate the median value for each marker and time point. Data analysis was done using non-parametric Wilcoxon matched-paired rank test with significant p values indicated. Baseline represents values prior to initiation of standard anti-TB drugs. 2nd and 6th month refers to time after therapy initiation. QFT⁺- Quantiferon TB gold assay positive for apparently healthy study participants.

negative PTB patients did not differ significantly from QFT positive study participants after six months of anti-TB drug treatment. On the contrary, smear positive and negative PTB exhibited significantly higher level of Ki-67 expression on IFN-γ⁺CD4⁺T cells at the six-month treatment cohort than QFT positive study participants (Fig 5A–5D, P < 0.005).

## Discussion

There has been progress in improving the sensitivity and specificity of tuberculosis (TB) diagnostic tools, which could help to prevent TB transmission in the community as well as in endemic countries' overcrowded clinical settings. However, the TB control strategy, which is primarily reliant on the DOTS program's performance, is hampered by the lack of sensitive and specific anti-TB treatment prediction and monitoring tools at the early therapeutics as well as end of treatment completion phase. The inadequacy of recently approved molecular diagnostic methods to distinguish live/dead bacill [2, 5, 7, 9–11], the delayed sputum culture turnaround time[1, 2, 6], as well as poor sputum quality after two months of anti-TB medication challenges the TB control program [2–5].

Biomarkers that can predict and monitor anti-TB treatment response in cases with paucibacillary TB, such as extrapulmonary TB or smear negative pulmonary TB, are, unfortunately,

scarce. Accordingly, we assessed these flow cytometric markers in monitoring and predicting the standard anti-TB treatment response of smear negative pulmonary TB, which is a major public health problem in TB and HIV-endemic areas such as Ethiopia.

In our study both smear positive and smear negative TB patients underwent very significant decreases in the expression of HLA-DR and CD38 among unstimulated cells. Because the frequency of T cells expressing these markers was substantially higher than that of T cells responding to PPD, it can be reasonably assumed that the vast majority of these cells are not specific for TB. T cell stimulation in response to specific antigen is well described to result in the induction of markers for activation and proliferation which subsequently decay as the stimulus is removed [19–21]. The observation here that of the decrease in expression of such molecules over many months of therapy presumably reflects the gradual diminution of myco-bacterial load and disappearance of pro-inflammatory and/or presence of anti-inflammatory mediators[19–26, 30] Further research is needed to define non-antigen specific activation in human disease, its loss with disease resolution, and its association with other biomarkers.

Among the smear negative PTB patients the levels of PPD specific IFN-γ+CD4+Tcells decreased only after six-months and not after two months although the other functional markers (IFN-γ+TNF-α+CD4+Tcells, TNF-α+CD4+ Tells) had comparable levels of response across different time points in our cohort. On the contrary, among smear positive PTB, there was a significant reduction in functional markers in the six-month cohort when compared to the baseline and second month cohorts, but only the level of IFN-γ+TNF-α+CD4+Tcells in the second month cohort showed a significant increase when compared to baseline data. The difference in immune responsiveness during standard anti-TB treatment, particularly significant cytokine response during six-month follow-up among smear positive PTB but not for smear negative PTB patients, could be due to the presence of a high level of pre-activated T cells in active PTB patients, which could be related with bacterial load. Variable changes in the frequencies of TB specific cytokine producing T cells after therapy have been observed with some studies reporting increases and others decreases in frequencies [19–26, 30]. Importantly, the total numbers of TB specific T cells during disease represent the sum of those present in blood, which are readily accessible, and those from tissues, which are not. Total T cell number reflects several mechanisms including T cell generation, typically by proliferation, apoptosis and redistribution, but the dynamics of these processes are very difficult to assess in human disease. In our study, it is tempting to speculate that the drop in frequencies at the 6-month time point, especially true for smear positive PTB patients may primarily reflect an apoptostis-related contraction mechanism which has been observed in many infectious diseases but the definitive clarification of this, as well as comparison with other studies is difficult given the aforementioned limitations.

Of particular importance in this study was the evaluation of co-expression of activation/ proliferation markers among TB specific cytokine producing cells. Adekambi *et al.* first showed that these markers decayed with therapy among patients with smear positive TB [19]. In our study here, among smear positive TB, we did observe a significant decrease in co-expressing cells with therapy, but this occurred primarily after 6 months and not after two months. Moreover, among patients with smear negative TB, consistent decreases were not observed. Importantly, these latter results contrast sharply with the expression of HLA-DR and CD38 among unstimulated CD4 T cells which *decreased* substantially at 6 months among both smear positive and negative patients. The implication of these results is that in this study cohort, activation molecules on TB specific T cells decayed either more slowly than described in other cohorts among smear positive cases, or hardly at all among smear negative patients. Thus, at first glance one could argue that, at least in the Ethiopian context, quantitation of activation/proliferation marker expression by antigen specific cytokine producing cells by flow

cytometry may have little value in monitoring disease therapy, particularly in the early months of therapy. On the other hand, our results raise the possibility that, particularly in smear negative TB, T cells are being persistently and specifically activated by TB antigens at time points well beyond presumed clinical cure. This implies the presence of either dead or live and presumably sequestered mycobacteria, able to sustain T cell activation. Persistent antigen specific T cells are presumed to play a role in maintenance of latent TB, but in contrast, T cells from latently infected individuals express low levels of activation molecules on TB specific cells, as demonstrated in many studies including our own [25, 31–33]. The clinical significance of persistent activation molecules on TB specific cells after therapy is not clear. It may be that this observation reflects therapy which was sufficient for apparent mycobacterial clearance using conventional sputum-based assays, but less optimal using potentially more sensitive assays. It will be important in future studies to determine whether or not there is a relationship between persistently activated T cells and risks for subsequent complications.

A limitation of our study is that we used PPD as a stimulus rather than more specific stimuli such as ESAT/CFP10 peptides. It is clear that the findings of this study will require confirmation using more specific T cell stimulation approaches (for example using ESAT/CFP10 peptides).

In conclusion, in this study we monitored patients with either smear positive or smear negative TB after therapy for the presence of activated TB specific cytokine producing CD4 T cells. While we did observe an expected decay in the expression of activation molecules among smear positive patients, this occurred only after 6 months of therapy; moreover, decay of activation molecules with therapy was less apparent among smear negative TB. These results imply that, at least in the Ethiopian setting, persistent activation among antigen specific TB specific cells may be occurring, even in the face of presumed curative therapy. Because there are very few studies on TB biomarkers among smear negative TB patients in particular, we recommend to expanding the research to include extra pulmonary TB patients, in those who are HIV positive and negative.

## Supporting information

**S1 Data.**
(RAR)

**S1 File.**
(DOCX)

**S1 Fig.**
(RAR)

## Acknowledgments

We would like to express our gratitude to the volunteers and study nurses who took part in this study.

## Author Contributions

**Conceptualization:** Jyothi Rengarajan, Liya Wassie, Rawleigh Howe.

**Data curation:** Ahmed Esmael, Adane Mihret.

**Formal analysis:** Ahmed Esmael, Adane Mihret, Tamrat Abebe, Joel Ernst.

**Funding acquisition:** Ahmed Esmael.

**Investigation:** Ahmed Esmael, Tamrat Abebe, Daniel Mussa, Sebsibe Neway, Joel Ernst, Jyothi Rengarajan, Liya Wassie, Rawleigh Howe.

**Methodology:** Ahmed Esmael, Adane Mihret, Tamrat Abebe, Daniel Mussa, Sebsibe Neway, Joel Ernst, Jyothi Rengarajan, Liya Wassie, Rawleigh Howe.

**Supervision:** Adane Mihret, Tamrat Abebe, Joel Ernst, Jyothi Rengarajan, Liya Wassie, Rawleigh Howe.

**Validation:** Rawleigh Howe.

**Visualization:** Ahmed Esmael, Rawleigh Howe.

**Writing – original draft:** Ahmed Esmael, Rawleigh Howe.

**Writing – review & editing:** Adane Mihret, Tamrat Abebe, Joel Ernst, Jyothi Rengarajan, Liya Wassie.

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
