## [Decision Letter · Decision Letter 0]

21 Mar 2022

PONE-D-22-05355Persistent expression of activation markers on Mycobacterium tuberculosis-specific CD4 T cells in smear negative TB patientsPLOS ONE

Dear Dr. Esmael,

Thank you for submitting your manuscript to PLOS ONE. After careful consideration, we feel that it has merit but does not fully meet PLOS ONE’s publication criteria as it currently stands. Therefore, we invite you to submit a revised version of the manuscript that addresses the points raised during the review process.

Please submit your revised manuscript. If you will need significantly more time to complete your revisions, please reply to this message or contact the journal office at plosone@plos.org. Please include the following items when submitting your revised manuscript:A rebuttal letter that responds to each point raised by the academic editor and reviewer(s). You should upload this letter as a separate file labeled 'Response to Reviewers'.A marked-up copy of your manuscript that highlights changes made to the original version. You should upload this as a separate file labeled 'Revised Manuscript with Track Changes'.An unmarked version of your revised paper without tracked changes. You should upload this as a separate file labeled 'Manuscript'.

We look forward to receiving your revised manuscript.

Kind regards,

Frederick Quinn

Academic Editor

PLOS ONE

Journal Requirements:

"NIH/Fogarty International Center Global Infectious Diseases grant D43TW009127

AHRI"

4. Thank you for stating the following in the Funding Section of your manuscript: 

"This work was supported in part by the NIH/Fogarty International Center Global Infectious Diseases grant D43TW009127 and core fund from Armauer Hansen Research Institute (AHRI). The funders had no direct role in the study design, data collection or analysis."

"NIH/Fogarty International Center Global Infectious Diseases grant D43TW009127

AHRI"

Reviewers' comments:

Reviewer's Responses to Questions

**Comments to the Author**

1. Is the manuscript technically sound, and do the data support the conclusions?

Reviewer #1: Yes

Reviewer #2: Partly

2. Has the statistical analysis been performed appropriately and rigorously? 

Reviewer #1: Yes

Reviewer #2: No

3. Have the authors made all data underlying the findings in their manuscript fully available?

Reviewer #1: Yes

Reviewer #2: Yes

4. Is the manuscript presented in an intelligible fashion and written in standard English?

Reviewer #1: Yes

Reviewer #2: No

5. Review Comments to the Author

Reviewer #1: This work sought to evaluate the activation/proliferation/functional markers of T cells in predicting and monitoring anti-TB response in smear-negative TB. They used polychromatic flow cytometry to assess selected PPT-specific T cell markers using fresh PBMC of smear-negative pulmonary TB patients. They found that levels of activation and proliferation among total unstimulated CD4 T cells decreased after therapy. Thus, activation parameters decline with therapy among TB-specific cytokine producing cells.

I have several minor comments for this work:

1. Several figures, which are pushing the same points, may be combined together so that there are fewer figures in the whole manuscript. For example, Figure 8 and 9 can be combined together.

2. Authors observed and analyzed most of phenotypes and PPD-specific effectors of CD4+T cells, but less focus on CD8 T cells. Can authors comparatively discuss the different roles of CD4 and CD8 T cells in smear-negative patients upon therapy?

3. There are several (or a few) papers in this field to study the kinetics of CD4+/CD8+T cells in patients in other regions before and after TB therapy. Authors should cite these works.

4. PPD is a good choice of stimulator for enhancing the activation of T cells and productions of cytokines by T cells. However, PPD is also a little bit weak, compared with other stimulators (e.g. Mtb lysates). Thus, authors should discuss a little bit about the selections of stimulators.

5. Usually, Golgi blocker should be included during the culture and stimulation of PBMC, but they didn’t mention that.

6. There is an additional “t” in the end of Figure 2 legends

Reviewer #2: In the submitted manuscript, the authors declared that T cell activation (HLA-DR, CD-38), proliferation (KI-67), and functional (IFN-γ, TNF-α) markers can be used as diagnostic markers in predicting and monitoring anti-TB responses in smear negative TB patients after curative therapy. However, I thought there were lacks of sufficient evidence to support this view in the article. And the writing and integration of the article was not complete enough to contribute. It is suggested that this article will be modified from the following aspects.

Major comments

1. The subject of the article was not clear, and lacked novelty. If the author thought the activated markers (HLA-DR, CD-38, KI-67 et al.) in CD4+T cells can be used as diagnostic markers in smear negative TB patients, please give more and enough evidence to prove this point.

2. Inappropriate descriptions and summaries. In the abstract section, I didn’t understand the meanings of the sentence “whereas a consistent decline in these markers was less apparent among smear negative PTB patients at the sixth month” in the result. Otherwise, it should summarize new conclusions or ideas in the conclusion section, not repeat previous research results. In sum, the sentences in the abstract section were not concise enough to elaborate the viewpoints of this paper.

3. Summary sentences in each paragraph did not clarify the main idea of the paragraph.

4. The image resolution was insufficient for Figure 1. The flow diagrams on figure1 lacked horizontal and vertical coordinates, and the specific loop-door mode was not clear. In addition, Figure 1-3 can be combined into a figure, and these figures were not arranged neatly. The annotation of the graph was not standard, and the specific sample number and statistical method used should be explained clearly

5. The description was not precise and professional.

6. Lack of the summary of the take-home message.

6. PLOS authors have the option to publish the peer review history of their article (what does this mean?). If published, this will include your full peer review and any attached files.

Reviewer #1: No

Reviewer #2: No

---

## [Author Response · Author response to Decision Letter 0]

12 May 2022

Response to Journal Requirements and reviewers 

Title: Persistent expression of activation markers on Mycobacterium tuberculosis-specific CD4 T cells in smear negative TB patients 

PONE-D-22-05355

Ahmed Esmael, Adane Mihret, Tamrat Abebe, Daniel Mussa , Sebsibe Neway, Joel Ernst, Jyothi Rengarajan, Liya Wassie, Rawleigh Howe

We would like to express our gratitude to both reviewers and editors for providing us with helpful feedbacks that helped us to enhance our paper so that it could be published in reputable journal. Based on the reviewers and journal requirements we modify our manuscript and cover page section. We responded to comments as follows:

A. Journal Requirements

Response: In accordance with the guiding line, we have made changes to the manuscript.

Response: The online financing information has been corrected, and the financial disclosure statement has been included to the cover page section.

3. Thank you for stating the following financial disclosure

Response: The online financing information has been corrected, and the financial disclosure statement has been included to the cover page section.

4. Thank you for stating the following in the Funding Section of your manuscript

Response : We have made correction with regarding to the funding section in the cover page section. 

5. 5. In your Data Availability statement, you have not specified where the minimal data set underlying the results described in your manuscript can be found. PLOS defines a study's minimal data set as the underlying data used to reach the conclusions drawn in the manuscript and any additional data required to replicate the reported study findings in their entirety. All PLOS journals require that the minimal data set be made fully available. For more information about our data policy, please see

Response: We have made correction with regarding to the data availability on the online version. 

6. Your ethics statement should only appear in the Methods section of your manuscript. If your ethics statement is written in any section besides the Methods, please move it to the Methods section and delete it from any other section. Please ensure that your ethics statement is included in your manuscript, as the ethics statement entered into the online submission form will not be published alongside your manuscript

Response: The ethical statement was only mentioned in the method section of the manuscript. 

B. Reviewer 1

1. Several figures, which are pushing the same points, may be combined together so that there are fewer figures in the whole manuscript. For example, Figure 8 and 9 can be combined together.

 Response: We have made modification based on the comments and combined in , the file already attached as figure 5. 

2. Authors observed and analyzed most of phenotypes and PPD-specific effectors of CD4+T cells, but less focus on CD8 T cells. Can authors comparatively discuss the different roles of CD4 and CD8 T cells in smear-negative patients upon therapy?

Response: We have already incorporated in the main manuscript, Introduction part, page 3. Since the level of cytokine expression from PPD specific CD8+ T cell in this study was very low, we restrict our analysis on CD4+ T cells. 

3. There are several (or a few) papers in this field to study the kinetics of CD4+/CD8+T cells in patients in other regions before and after TB therapy. Authors should cite these works

Response: We have acknowledge the scientific works of many authors in our manuscript , typically in the introductory and discussion parts. 

4. PPD is a good choice of stimulator for enhancing the activation of T cells and productions of cytokines by T cells. However, PPD is also a little bit weak, compared with other stimulators (e.g. Mtb lysates). Thus, authors should discuss a little bit about the selections of stimulators

Response: We have already incorporated in the main manuscript, introduction section, page 3.

 5. Usually, Golgi blocker should be included during the culture and stimulation of PBMC, but they didn’t mention that.

Response: We have already incorporated in the main manuscript, method part, page 5

6. There is an additional “t” in the end of Figure 2 legends

Response: I have already incorporated in the main manuscript, figure 2, page 9

C. Reviewer 2

1. The subject of the article was not clear, and lacked novelty. If the author thought the activated markers (HLA-DR, CD-38, KI-67 et al.) in CD4+T cells can be used as diagnostic markers in smear negative TB patients, please give more and enough evidence to prove this point

Response: In the methods section, we tried to elaborate about the study subjects (page 4). 

Normally, the main objective of this study was to evaluate the potential of activation and proliferation markers to predict and monitor anti-TB treatment response in smear negative pulmonary TB patients. While many studies have shown that the level of activation and proliferation markers could be used to predict and monitor anti-TB treatment response in smear positive pulmonary TB patients, however those markers not explored in depth in smear negative pulmonary TB patients. As far as I know, no previous study has evaluated the potential of such biomarkers to predict and monitor anti-TB treatment response in smear negative pulmonary TB patients which is prevalent in resource limited sub-Saharan Africa including Ethiopia. As we mentioned in the results section, high levels of activation markers expression in smear negative pulmonary TB patients persisted even after treatment was completed and patients were clinically resolved, implying that activation and proliferation markers are unsuitable for predicting and monitoring anti-TB drug treatment response in smear negative pulmonary TB patients in Ethiopian context.

2. Inappropriate descriptions and summaries. In the abstract section, I didn’t understand the meanings of the sentence “whereas a consistent decline in these markers was less apparent among smear negative PTB patients at the sixth month” in the result. Otherwise, it should summarize new conclusions or ideas in the conclusion section, not repeat previous research results. In sum, the sentences in the abstract section were not concise enough to elaborate the viewpoints of this paper.

 Response: Please deeply apologies for the ambiguous statement regarding the less consistent decrease in activation marker expression on antigen specific CD4 T cells in smear negative TB patients. We used this statement to refer, when we compared the level of activation markers in smear positive and smear negative PTB patients after six months of standard anti-TB treatment, the level of activation in smear positive PTB patients decreased significantly, but the level of activation in smear negative PTB patients remained high, and we reasoned that smear negative PTB patients had a persistent level of activation. In the abstract section, we also incorporated a new concluding sentence. 

3. Summary sentences in each paragraph did not clarify the main idea of the paragraph

Response: We reviewed the entire manuscript and made modifications as needed, explicitly displaying the changes we made with track changes.

4. The image resolution was insufficient for Figure 1. The flow diagrams on figure1 lacked horizontal and vertical coordinates, and the specific loop-door mode was not clear. In addition, Figure 1-3 can be combined into a figure, and these figures were not arranged neatly. The annotation of the graph was not standard, and the specific sample number and statistical method used should be explained clearly

Response: We have incorporated the comments in the main manuscript, result section, page 9. 

5. The description was not precise and professional.

Response: We reviewed the entire manuscript and made modifications as needed, explicitly displaying the changes we made with track changes.

6. Lack of the summary of the take-home message

Response: We have incorporated the comments in the main manuscript, discussion section, page 23 

With regards

Ahmed Esmael

---

## [Decision Letter · Decision Letter 1]

30 May 2022

PONE-D-22-05355R1Persistent expression of activation markers on Mycobacterium tuberculosis-specific CD4 T cells in smear negative TB patientsPLOS ONE

Dear Dr. Esmael,

Thank you for submitting your manuscript to PLOS ONE. After careful consideration, we feel that it has merit but does not fully meet PLOS ONE’s publication criteria as it currently stands. Therefore, we invite you to submit a revised version of the manuscript that addresses the points raised during the review process.

Please submit your revised manuscript by Jul 14 2022 11:59PM. If you will need significantly more time to complete your revisions, please reply to this message or contact the journal office at plosone@plos.org. Please include the following items when submitting your revised manuscript:A rebuttal letter that responds to each point raised by the academic editor and reviewer(s). You should upload this letter as a separate file labeled 'Response to Reviewers'.A marked-up copy of your manuscript that highlights changes made to the original version. You should upload this as a separate file labeled 'Revised Manuscript with Track Changes'.An unmarked version of your revised paper without tracked changes. You should upload this as a separate file labeled 'Manuscript'.If applicable, we recommend that you deposit your laboratory protocols in protocols.io to enhance the reproducibility of your results. Protocols.io assigns your protocol its own identifier (DOI) so that it can be cited independently in the future. For instructions see: https://journals.plos.org/plosone/s/submission-guidelines#loc-laboratory-protocols. Additionally, PLOS ONE offers an option for publishing peer-reviewed Lab Protocol articles, which describe protocols hosted on protocols.io. Read more information on sharing protocols at https://plos.org/protocols?utm_medium=editorial-email&utm_source=authorletters&utm_campaign=protocols.

We look forward to receiving your revised manuscript.

Kind regards,

Frederick Quinn

Academic Editor

PLOS ONE

Journal Requirements:

Reviewers' comments:

Reviewer's Responses to Questions

**Comments to the Author**

1. If the authors have adequately addressed your comments raised in a previous round of review and you feel that this manuscript is now acceptable for publication, you may indicate that here to bypass the “Comments to the Author” section, enter your conflict of interest statement in the “Confidential to Editor” section, and submit your "Accept" recommendation.

Reviewer #1: All comments have been addressed

Reviewer #2: All comments have been addressed

2. Is the manuscript technically sound, and do the data support the conclusions?

Reviewer #1: Partly

Reviewer #2: Partly

3. Has the statistical analysis been performed appropriately and rigorously? 

Reviewer #1: Yes

Reviewer #2: (No Response)

4. Have the authors made all data underlying the findings in their manuscript fully available?

Reviewer #1: Yes

Reviewer #2: Yes

5. Is the manuscript presented in an intelligible fashion and written in standard English?

Reviewer #1: Yes

Reviewer #2: Yes

6. Review Comments to the Author

Reviewer #1: They have addressed most of my comments. They may need to include more references (regarding immune responses CD4+or CD8+ T cells before and after anti-TB therapeutics in TB patients) from other labs in the similar field. I have no more other comments.

Reviewer #2: 1.The revised version of the article is much better.

2.The arrangement of graphs is not very orderly, and the font size is inconsistent. Please readjust them.

3.Please simplify the discussion section according to the results and the main idea of the article.

7. PLOS authors have the option to publish the peer review history of their article (what does this mean?). If published, this will include your full peer review and any attached files.

Reviewer #1: No

Reviewer #2: No

---

## [Author Response · Author response to Decision Letter 1]

6 Jun 2022

Response to editors requirements 

Title: Persistent expression of activation markers on Mycobacterium tuberculosis-specific CD4 T cells in smear negative TB patients 

PONE-D-22-05355

Ahmed Esmael, Adane Mihret, Tamrat Abebe, Daniel Mussa , Sebsibe Neway, Joel Ernst, Jyothi Rengarajan, Liya Wassie, Rawleigh Howe

We would like to express our gratitude to editors for providing us with helpful feedbacks that helped us to enhance our paper so that it could be published in reputable journal. Based on the reviewers and journal requirements we modify our manuscript and cover page section. We responded to comments as follows:

A. Journal Requirements

1. Please review your reference list to ensure that it is complete and correct.

Response: Thank you for your comments. Deeply apology for the citation problems in the discussion part. I do not know why such issues as far as I used endnote reference citation tool. I made revision for some references which needs some correction, especially related with formatting, completeness and citation issues in the discussion part. 

B. Response for Reviewer #1

1. They may need to include more references (regarding immune responses CD4+or CD8+ T cells before and after anti-TB therapeutics in TB patients) from other labs in the similar field. I have no more other comments.

Response: We incorporated and acknowledged similar works in the introductory part. 

C. Response for Reviewer #2

1. The arrangement of graphs is not very orderly, and the font size is inconsistent. Please readjust them.

Response: We made modification for font size and arrangement for all graphs in the manuscript and tried to re-organize the order in the result section. 

2. Please simplify the discussion section according to the results and the main idea of the article.

Response: We made revision in the discussion part without affecting the main finding. 

 With regards

Ahmed Esmael

---

## [Decision Letter · Decision Letter 2]

27 Jun 2022

Persistent expression of activation markers on Mycobacterium tuberculosis-specific CD4 T cells in smear negative TB patients

PONE-D-22-05355R2

Dear Dr. Esmael,

We’re pleased to inform you that your manuscript has been judged scientifically suitable for publication and will be formally accepted for publication once it meets all outstanding technical requirements.

Kind regards,

Frederick Quinn

Academic Editor

PLOS ONE

Additional Editor Comments (optional):

Reviewers' comments:

Reviewer's Responses to Questions

**Comments to the Author**

1. If the authors have adequately addressed your comments raised in a previous round of review and you feel that this manuscript is now acceptable for publication, you may indicate that here to bypass the “Comments to the Author” section, enter your conflict of interest statement in the “Confidential to Editor” section, and submit your "Accept" recommendation.

Reviewer #1: All comments have been addressed

Reviewer #2: All comments have been addressed

2. Is the manuscript technically sound, and do the data support the conclusions?

Reviewer #1: Partly

Reviewer #2: Yes

3. Has the statistical analysis been performed appropriately and rigorously? 

Reviewer #1: N/A

Reviewer #2: Yes

4. Have the authors made all data underlying the findings in their manuscript fully available?

Reviewer #1: Yes

Reviewer #2: Yes

5. Is the manuscript presented in an intelligible fashion and written in standard English?

Reviewer #1: Yes

Reviewer #2: Yes

6. Review Comments to the Author

Reviewer #1: They have addressed most of my previous comments. But I suggest them to use Endnote to prepare the references in the manuscript.

Reviewer #2: (No Response)

7. PLOS authors have the option to publish the peer review history of their article (what does this mean?). If published, this will include your full peer review and any attached files.

Reviewer #1: No

Reviewer #2: No

---

## [Editor Report · Acceptance letter]

29 Jul 2022

PONE-D-22-05355R2 

Persistent expression of activation markers on *Mycobacterium tuberculosis*-specific CD4 T cells in smear negative TB patients 

Dear Dr. Esmael:

I'm pleased to inform you that your manuscript has been deemed suitable for publication in PLOS ONE. Congratulations! Your manuscript is now with our production department. 

Kind regards, 

on behalf of

Dr. Frederick Quinn 

Academic Editor

PLOS ONE